# Female Sexual Violence: A 12-Year Experience at a Single University Hospital in North-East Italy

**DOI:** 10.3390/ijerph21030301

**Published:** 2024-03-05

**Authors:** Serena Xodo, Veronica Tius, Giovanni Baccarini, Lorenza Driul

**Affiliations:** 1Clinic of Obstetrics and Gynecology of Udine, Azienda Sanitaria Universitaria Friuli Centrale, Piazzale Santa Maria della Misericordia 15, 33100 Udine, Italy; giovanni.baccarini@asufc.sanita.fvg.it (G.B.); lorenza.driul@uniud.it (L.D.); 2Department of Gynecology and Obstetrics, School of Medicine of Udine, 33100 Udine, Italy; veronica.tius@gmail.com; 3Dipartimento di Area Medica, Università degli Studi di Udine, 33100 Udine, Italy

**Keywords:** sexual violence, physical violence, abuse

## Abstract

This retrospective study analyzed a case series of female sexual violence (SV) victims who were admitted to the emergency department of the University Hospital in Udine between January 2012 and April 2023. A total of 155 cases were divided into two groups according to their age: 115 adult victims and 40 minors. Compared with minors, adults had risk factors such as psychiatric disorders and past experience of SV, and reported bodily injuries and extragenital lesions more frequently. Moreover, a positive screening for sexually transmitted diseases and its association with genital injuries turned out to be significantly more present among adult victims than minors. In contrast, victims younger than 18 years tended to delay seeking medical help and more often did not report genital penetration. To conclude, a deeper knowledge of the different characteristics of sexual abuse among female adults and minors may help us to understand what the focus of prevention programs and public awareness campaigns should be.

## 1. Introduction

We are currently living in a time of strong political momentum that has raised awareness of violence against women and girls in health agendas. This global trend provides a unique opportunity to strengthen the response to gender-based violence within the health system. Violence against women includes different forms of violence, such as violence by intimate partners or by family members, sexual violence (SV) by non-partners, trafficking for sexual and economic exploitation, femicide, acid throwing, sexual harassment in schools, workplaces and public places, and also online harassment through the Internet or social media. Moreover, specific types of violence originate from socially accepted gender discrimination, giving rise to shame and stigma being experienced by women and girls, who therefore tend to keep the violence hidden [1]. Similarly, violence against children (aged 0–18 years) is widespread, often invisible and has long-term adverse consequences, such as health risk behaviors and the tendency to experience and perpetrate subsequent violence. Girls face all forms of child maltreatment, as well as specific forms of gender-based violence, such as female genital mutilation or early and forced marriage. Additionally, girls are more likely to experience sexual abuse or be trafficked for sex than boys, and adolescent girls are more likely to experience intimate partner violence (IPV).

Sexual violence is inherently rooted in the lesser power that women have in society compared to men [2]. From this point of view, rape may be an act of punishment and an expression of holding power over a victim [3,4]. Various risk factors have been associated with the perpetration of rape. These include adverse childhood experiences, attachment and personality disorders, social learning and delinquency, adherence to gender-inequitable masculinities, and substance abuse [1]. Recent research has confirmed that alcohol plays an important role in a high proportion of rape cases. It not only acts as a situational factor, but has the pharmacological ability to reduce inhibitions. Consequently, individuals under the influence of alcohol may feel less accountable for their actions, contributing to instances of SV, especially in group settings where alcohol consumption is prevalent [5,6]. Of particular note is the strong correlation between all forms of child abuse and the subsequent perpetration of rape [3]. Additionally, poverty exacerbates the likelihood of adverse childhood experiences and involvement in antisocial peer groups such as gangs, thereby increasing the risk of engaging in sexually violent behavior. It is important to recognize that the perpetration of rape is highly gendered, being closely related with societal constructions of masculinity that emphasize male dominance and control over women [1]. Addressing SV necessitates dismantling these entrenched gender norms and power dynamics.

Intimate SV is common and occurs in all socioeconomic backgrounds and cultural groups [7]. The factors contributing to the occurrence of IPV can be categorized into individual, relationship, community, and societal domains. When examining individual risk factors, aspects such as young age, limited educational attainment, exposure to or experience of violence during childhood, substance abuse, personality disorders, acceptance of violence, and a history of past abusive behavior toward partners have been identified as significant contributors. In the realm of relationship risk factors, conflict or dissatisfaction within the relationship, economic stress, exposure to parental violence during childhood, and educational disparities play pivotal roles in amplifying the likelihood of IPV. Community and societal factors refer to gender-inequitable social norms, the limited civil rights of women, weak community sanctions against IPV, pervasive poverty, the widespread acceptance of violence as a conflict resolution method, armed conflicts, and high societal violence levels [1,7].

Child abuse tends to occur in families who experience a combination of several risk factors [8]. These encompass socioeconomic marginalization indicators, including poverty, unemployment, and limited educational opportunities. Within the family dynamics, strained interactions marked by violence among members or serious marital discord extend their impact to parenting responsibilities. Parental dysfunction becomes a contributing factor, involving issues related to mental health, behavior, or medical conditions that affect their ability to provide adequate care. Additionally, exposure to neighborhood violence is prevalent, creating unsafe environments for children and fostering a societal acceptance of physical punishment or violence directed towards them. Together, these factors form a complex web leading to an increased risk of child abuse within vulnerable families. Recent research has shed light on the prevalence of severe stressors within families being reported to influence child maltreatment. Approximately half of these families grapple with a combination of significant stressors, including unemployment, substance abuse, poverty, and exposure to neighborhood violence. The convergence of these risk factors heightens the likelihood of one or more children being removed from the home, emphasizing the critical need for targeted intervention and support in such vulnerable family contexts [9].

Understanding the factors underlying rape perpetration, domestic violence and child abuse is a fundamental building block for effective prevention and health care programs. In Italy, the national guidelines for a correct response to the physical and psychological consequences produced by violence against women were adopted in November 2017 [10]. These guidelines outline a path that accompanies abused women, from their request for medical help to their discharge from dedicated public or private institutions. The number of women requiring medical help because of violence in Italy steadily increased over the period 2017–2019, before the pandemic outbreak. According to the Italian national statistical institute, this number rose from 14,368 in 2017 to 15,800 in 2019. During the pandemic, the number of women requiring medical help for violence significantly declined in Italy, and this tendency was also observed globally. In the following years (2021 and 2022), a greater diffusion of violence against women was reported in our country, with 12,780 and 14,448 women accessing emergency care, respectively. Those most affected were foreign and young women, aged 18–34 years. The most common forms of violence against girls aged less than 18 years are sexual abuse and child maltreatment, and against adult women they are adult maltreatment, post-traumatic stress disorder and sexual abuse [11]. 

The aim of this study was to illustrate the features of a series of female patients requiring help for SV at the university hospital of Udine over the last 12 years, focusing on the differences in characteristics, clinical findings and management of minors and adults.

## 2. Materials and Methods

This retrospective study was approved by the institutional review board (Prot. IRB: 262/2023). The study included all patients admitted to the emergency department of the University Hospital of Udine between January 2012 and April 2023 because of sexual assault. The emergency department of Udine serves a catchment area of about 500,000 people located in the middle area of Friuli Venezia Giulia, and is both a primary care facility (walk-in patients) and a tertiary referral center for other minor hospitals in the area. Cases were retrieved from the electronic patient chart database using the terms “sexual violence” or “sexual assault” for a comprehensive full-text search. The charts for all cases were reviewed; only female cases were included. A violence offence was defined as “any sexual act, attempt to obtain a sexual act, unwanted sexual comments or advances, or acts to traffic or otherwise directed against a person’s sexuality using coercion, by any person regardless of their relationship to the victim, in any setting, including but not limited to home and work” [1]. Data extraction was performed by one of the authors (VT) for the entire study. All patients were managed according to the institutional protocol, which has been revised over time. Its last version was released in 2018, following the national guidelines on SV published in 2017; these aimed to improve service infrastructure, referrals, and the accessibility, availability and quality of care, and to train the health workforce. In our hospital, the first clinical examination is carried out by gynecologists in case of SV on female adults and minors. Our approach to addressing cases of SV entails gathering a comprehensive range of data concerning each survivor. This includes details about their socio-demographic background, medical history, the nature of the abuse they experienced, and the collection of various forms of medical evidence. Specifically, we conduct tests to detect infectious diseases, analyze blood and urine for toxicological substances, and obtain swabs to capture biological traces. All of this pertinent information is meticulously documented within the individual’s medical records. According to our protocol, the victim is offered post-coital contraception, antibiotic therapy for the prevention of sexually transmitted diseases and psychological support.

The following data were collected for each patient: age; vulnerability factors for SV, such as a history of previous SV and/or maltreatment, alcohol or substance abuse, the presence of mental disorders or other disabilities, a history of custody or stay in the community, and a history of prostitution; the characteristics of the perpetrator (in particular, the relationship with the victim); the interval between the abuse and accessing the hospital for medical evaluation; the presence of genital and/or extragenital injuries; and tests for sexually transmitted diseases (STDs), spermatozoa research, as well as toxicological screening. Additionally, data were gathered on the administration of antibiotics to prevent STDs, HIV prophylaxis, and emergency contraception. Patients were considered affected by a mental health disorder when carrying a specific diagnosis given by a psychiatric specialist according to the fifth edition of the Diagnostic and Statistical Manual of Mental Disorders (DSM-5). Generally, patients with this kind of diagnosis were followed by a Mental Health Center or by general practitioners. Anxiety and depression were among the most common types of mental health conditions that our patients experienced. Additionally, genital penetration was reported when declared by the victim and then compared with medical findings.

For the present research, the data obtained from consulting medical records were reported in a digital data set and subsequently analyzed with descriptive statistics. To better describe the phenomenon in our female population, we divided our case series into two groups according to their age: the first encompassing adult victims, older than 18 years, the second including minors, younger than 18 years. The statistical analysis was carried out using GraphPad Prism software (version 9.5.1). Continuous variables were reported as means with their standard deviations, and qualitative variables were reported as frequencies and percentages. The significance of comparisons was assessed using the independent Student *t* test for quantitative variables and using the Chi square test for qualitative variables. A *p*-value < 0.05 was considered statistically significant.

## 3. Results

From January 2012 to April 2023, a total of 159 cases of SV referred to our department were selected. Given that a highly trained health care team is required to correctly use the institutional protocol, especially during the collection of biological samples, the personnel of our department was also employed in case sexual abuse against male patients was reported after 2018. Therefore, two cases were ruled out because they represented male victims. Another two cases were then excluded because the victims were in confusion and not sure whether or not the sexual assault had occurred. These victims decided to not proceed further with the medical evaluation. Hence, 155 cases were finally included in our analysis, and divided into 2 groups according to their age: 115 were adult victims and formed the adult group, whereas 40 were younger than 18 years and represented the minor group.

### Results and Tables

Overall, our case series had a mean age of 28.48 (+15.36). Most victims were of Italian nationality (61.2%) and single (79.3%). Almost one in three patients were unemployed (24.5%) and nearly half of the victims (47.7%) had a history of previous SV.

Table 1 describes the socio-demographical characteristics and risk factors for SV in the general population and in the minor group versus the adults group. The mean age in the minor group was 10.8 (+4.94), while the mean age in the adult group was 34.6 (+12.8); most victims in both groups were Italian (67.5% vs. 59.1%). Previous experience of SV and mental health disorders were reported more frequently among adults than minors (previous SV: 30% vs. 53.9%; *p* value: 0.010; mental health disorder: 5.0% vs. 25.2%; *p* value: 0.005).

Appendix A describes the characteristics of sexual abuse and the reported gynecological or physical injuries of all cases. Sexual violence was disclosed in 93.5% of cases, a gang rape was disclosed in 6.4% of cases and physical violence was experienced by 12.9% of patients. Of note, in nearly one third of cases (24.5%), penetration could not be determined for sure. To better describe the problem, we recall six interesting cases.

Case number 1 was a 4-year-old girl, accompanied by her mother to a pediatric consultation because of anal pain. The child had repeatedly indicated that her 12-year-old male cousin was involved. However, the penetration could not be determined for certain since the victim was not able to express herself in clear terms. Of note, the child reported hyperemia of the labia minora and perianal area, but the medical evaluation happened more than 24 h after the suspected abuse. 

Case number 2 was a 2-year-old girl accompanied by her mother to a pediatric consultation because of suspected sexual abuse following the irritability of her daughter when she was with her stepfather. Genital penetration could not be determined, since the little girl did not disclose it explicitly. The girl had hyperemia of the external genitalia, but the suspected abuse could not be located in time. 

Case number 3 was a 1-year-old baby, accompanied by her mother to a pediatric consultation because of suspected childhood abuse. The mother told clinicians that the baby cried whenever the father appeared and refused to sit on his lap. To obtain a deeper insight into the family situation, a medical report revealed that the parents were splitting up. Genital penetration could not be ascertained since the baby was clearly too immature to express herself. Moreover, the last time the baby was reported to be with her father was 2 months earlier. Interestingly, hyperemia of the labia majora was found in this case, too.

Case number 4 was a 17-year-old girl, who was found close to her school in an altered state of consciousness and with multiple bodily injuries (especially on her face) after having consumed alcohol and illicit substances. Given the extraordinary circumstances of her finding, sexual assault by a stranger was immediately suspected. However, the adolescent was not able to recall the event or penetration, and SV could not be determined, even though a medical evaluation was performed 24 h after the suspected abuse.

Similarly, case number 5 was a 14-year-old girl who was found almost naked in a public garden in an alcohol and drug-induced coma after attending a party with friends. She had multiple extragenital injuries, especially on her legs. The victim could not tell if SV had occurred, due to her altered state of consciousness, but it was suspected by parents and emergency doctors, given the circumstances. 

Case number 6 was a 4-year-old girl, accompanied by her mother to a pediatric consultation because of suspected abuse. They both lived in an anti-violence center. The little girl was observed to touch her own genitalia too often and when asked to identify who taught her such behavior, she indicated the parents as well as other relatives. The little girl did not have any genital or extragenital lesion and, even in this case, penetration could not be recognized. These examples illustrate how challenging it can be to conduct a medical evaluation of a sexually abused child, especially when physicians are unfamiliar with the wide variation in the normal genital anatomy of prepubertal girls.

In our case series, alcohol and/or drug abuse was disclosed by 24.5% of patients and most cases who did not remember the sexual abuse had a positive toxicological screening (70.5%). The aggressor was known in 65.8% of cases, and the time between the violent episode and the request for medical help ranged over 6 days. As for the medical examination, genital injuries were reported in 16.7% of cases, extragenital injuries were present in 41.2% of cases, and no injuries were detectable in 44.5% of cases. When STD tests were performed, the samples turned out to be positive in 45.6% of cases, detecting HPV infection in 77.7%, HCV in 4.7%, HSV1-2 in 9.5% and Chlamydia in 11.1% of cases. When a spermatozoa research test was carried out, sperm was retrieved in 27.7% of cases. More than half of the patients received the antibiotic prophylaxis against STDs, and slightly more than one third of patients received emergency contraception. Victims were discharged home in 50.3% of cases, to the emergency room in 27% of cases, home but with the support of a social worker in 10.3% of cases and finally to an anti-violence center in 3.2% of cases.

Table 2 describes the characteristics of sexual abuse and the reported gynecological or physical injuries of the minor group versus the adult group. The adult group turned out to be more frequently affected by physical violence (minors 0% vs. adults 17.3%; *p*-value: 0.0021) and specifically by extragenital lesions (minors 12.5% vs. adults 51.3%; *p*-value < 0.0001). In contrast, minors reported no injuries more frequently (minors 65% vs. adults 37.3%; *p*-value: 0.003). Women more frequently had a positive screening for STDs (minors 11.4% vs. adults 57.2%; *p*-value < 0.0001) and a positive screening in association with genital injuries (minors 0% vs. adults 53.3%; *p*-value: 0.019). Obviously, our analysis revealed that prophylaxes were more often administered in the adult group than in the minors one. To conclude, compared with older victims, those younger than 18 years took longer to seek medical help (minors 14.07 days vs. 3.7 days; *p*-value 0.006) and received psychological support more often (minors 55% vs. adults 17.3%; *p*-value < 0.0001).

## 4. Discussion

### 4.1. Principal Findings

Overall, this retrospective study shows that victims of SV in our geographical area have a mean age of 28 and that young women (<30 years) are involved the most. This study demonstrates that, compared with minors, adults have risk factors such as mental health disorders and past experience of SV, and report bodily injuries and extragenital lesions more frequently. Moreover, a positive screening for STDs and its association with genital injuries turned out to be significantly more present among adult victims than minors. By contrast, victims younger than 18 years tended to delay seeking medical help and more often did not report genital penetration.

### 4.2. Results in the Context of What Is Known

In our case series, women older than 18 years appeared to be significantly more affected by mental health disorders and had more often experienced SV in the past. In contrast, a history of alcohol and drug abuse and a history of prostitution were not commonly reported. Therefore, they were not significantly different between the groups. A systematic review conducted in 2020 revealed that nearly 40% of individuals seeking assistance at Sexual Assault Referral Centers in England were also receiving care from mental health services [12]. Similarly, a study conducted in the United States identified “major psychiatric diagnoses” in 26% of 819 female patients who presented to an Emergency Department following the report of sexual assault [13]. Moreover, various studies have highlighted the significant prevalence of victimization among individuals with mental illness, including instances of sexual assault [14,15]. These findings confirm that psychiatric conditions increase the vulnerability of women to SV and abuse. In individuals with major depressive disorder, the slowing of thoughts and movement, coupled with feelings of worthlessness and indecisiveness, can significantly impact their ability to perceive and respond to threatening situations. This compromised cognitive and emotional state may hinder the individual’s capacity to identify and appropriately address potential threats [16,17]. On the other hand, episodes of mania, characteristic of bipolar disorder, may manifest with elevated mood, impaired judgment, disinhibition, and hypersexuality, posing a risk of engaging in activities with potentially painful consequences, such as sexual incautions. An impaired perception of the surrounding environment and diminished social functioning during manic episodes may further contribute to difficulties in staying safe, potentially leading to misjudgments regarding trust and increasing vulnerability to violence. Individuals with a history of SV, whether as adults or children, are reportedly at risk of using sex as a form of self-harm, potentially leading to re-victimization. Furthermore, drug abuse can result in a reduced awareness of one’s surroundings, impairing one’s ability to recognize and respond to danger. In the context of personality disorders, varying types may be associated with impulsivity, diminished self-worth, and submissiveness. These characteristics could contribute to challenges in maintaining mindfulness and, consequently, in ensuring personal safety [13,16,17]. Experiences with violence shape human development. Our findings, together with data in the literature, may suggest that being violently victimized, or witnessing serious violence early in life, might increase the risk of victimization later in life because of altered psychological development. Nonetheless, the continuity of adversity throughout life might make early victimization, a marker of social disadvantage, more likely to reproduce also later in life.

Our study showed that adults had bodily injuries more often than minors. This observation could be explained by the fact that sexual assault occurs unpredictably in adult women, and that bodily injuries are the result of their physical resistance against the aggressor or of a deliberate measure by the assailant. Our speculation is further confirmed by the fact that the aggressor was more frequently known among minors than adults, even though this difference was not statistically significant. Interestingly, the perpetrator was a family member in most cases reported in the minor group. These data shed light on the type of SV occurring against minors in our case series, which was nearly always performed in the family environment. Remarkably, our study revealed that the time interval between the SV and the request for medical help (expressed in days) was significantly longer among minors than adults. It is possible that the relationship between the perpetrator and the victim, as well as the victim’s feelings towards the offender, typical features of the family milieu, may lead to a delayed disclosure of SV among young victims [18]. Previous research has indeed demonstrated that victims of intra-familial abuse are more likely to delay disclosure than victims of extra-familial abuse [19]. In instances where the perpetrator is a family member or a trusted adult, a young child might be coerced into silence through threats of punishment, manipulation by instilling a belief that nobody would believe them, or by suggesting that revealing the truth would lead to the abuser going to jail. On the other hand, an older child may internalize the abuse as their own fault, harboring feelings of embarrassment and shame that prevent them from confiding in anyone. It is not uncommon for an older sibling to disclose their abuse only upon discovering that a younger sibling is also being victimized, fueled by a desire to protect their sibling from the same harm. For young adolescents, the fear of disclosing abuse is heightened by the circumstances surrounding the incidents. Instances where the abuse occurred during engagement in high-risk behaviors, against explicit warnings, or in undisclosed and unsafe locations may induce feelings of shame, regret, and apprehension about parental reactions. Consequently, these emotions might lead to a significant delay in disclosing the abuse, sometimes spanning weeks or even months [18,20].

No significant difference was reported regarding the prevalence of genital lesions between the groups. Moreover, in two thirds of young victims, no injuries were detected, likely rejecting an alleged sexual offence. However, it could be hypothesized that the victim might develop a passive behavior toward the sexual offense, mostly if the abuse occurs in the family environment. In rare cases, during the abuse, the victim might develop a set of involuntary responses, including temporary muscular paralysis, lowered body temperature, uncontrollable tremors and analgesia, which is called tonic immobility. Tonic immobility is considered the last chance when other defense attempts have failed [21] and its occurrence during sexual abuse cannot be excluded in the most vulnerable victims, such as minors in our case series.

Another challenging problem faced by clinicians was the inability to determine whether or not penetration had occurred in some victims. This situation only involved children who could not yet speak or were not mature enough to give meaning to sexual actions, or, again, were in the middle of a conflict between parents who were in the process of separation. Some noteworthy observations could be drawn from the six ambiguous cases described in the results section.

First, it should be clarified what is meant by genital penetration. The female genital anatomy consists of external structures (labia majora, labia minora and the enclosed vestibule) and internal structures (hymen, vagina, uterus and adnexa). Penetration may be limited to the external genitalia. In this specific situation, it is unlikely that penetration will lead to physical signs, other than transient redness or abrasion, which heals fast. Second, it should be considered that, anatomically, the female genitalia, the anus and the mouth can be penetrated without any visible injuries, thus complicating the medico-legal evaluation. Therefore, the clinical assessment of children who may have been sexually abused can seldom confirm and never exclude sexual abuse. Third, even if false allegations might be commonly reported when a serious conflict exists between parents in the process of separation, the disclosure of sexual abuse against children should never be discouraged. A deferred admission of SV has multiple important negative consequences, such as an increased risk of severe psychological repercussion and a decreased ability to diagnose injuries and to detect biological traces of the violence. In a recent update on the classic interpretation of medical findings related to suspected child sexual abuse, Kellogg and colleagues have brought attention to the evolving understanding of normal genital anatomy [22]. Notably, S. Starling, in an editorial introducing Kellogg’s work, acknowledges that there has been a substantial shift in perspective since 1992, when an enlarged hymenal opening would lead to the suspicion of abuse [23]. Over the years, the definition of normal findings has expanded, while the list of trauma-induced findings has remained largely unchanged. Despite this evolution, certain aspects continue to spark debates among experts. One such point of contention is the interpretation of erythema. The editorial highlights the absence of a consensus on how to categorize erythema, noting that while the interpretation table designates erythema as a normal finding, widely used diagnostic criteria such as TEARS (Tears, Ecchymosis, Abrasions, Redness, and Swelling) consider it as indicative of trauma [24]. According to the TEARS criteria, the presence of redness or any other listed finding results in the examination being recorded as having a positive finding. Moreover, the authors suggest that very few findings are diagnostic of abuse: acute trauma to genital/anal tissues (such as acute laceration(s) or the bruising of different parts of the genitalia or perineum), residual injuries to genital/anal tissues (perianal scar, scar of posterior fourchette or fossa, hymenal transection or cleft, signs of female genital mutilation or cutting) and acute oral trauma (such as unexplained injury or petechiae of the lips or palate, particularly near the junction of the hard and soft palate). The identified findings strongly indicate potential abuse, even when there is no explicit disclosure from the child. However, an exception is made if the child or caretaker promptly provides a plausible explanation involving accidental anogenital straddle, crush, impalement injuries, or verified details of past surgical interventions from medical records. In cases where there are isolated, few, or superficial injuries resembling bruises or petechiae, it should be imperative to validate them as traumatic injuries by observing their resolution during follow-up examinations. To ensure precision in diagnosis, it is recommended that these findings are documented through photographs or video recordings. The evaluation and confirmation of these visual records should be conducted by an expert in sexual abuse assessment, emphasizing the importance of accuracy in the diagnostic process [22,25].

Importantly, an element that the evaluator must take into consideration is the mode in which the minor victim narrates the suspected abuse, especially if pre-adolescent or younger children are involved. Often, these narratives are inconsistent and characterized by bizarre, unusual, or “fantastic” situations that can undermine the credibility of the victim and pose challenges for clinical and forensic interpretation. Longobardi et al. attempted to classify recurring bizarre phrases in the accounts of victims of suspected sexual abuse in particular, highlighting how these are often present in the context of child pornography, and therefore may actually testify to genuinely experienced events. Consequently, it is crucial for the clinician to faithfully report the accounts of younger victims, providing detailed information, including any elements judged as strange, for subsequent evaluation by law enforcement, child neuropsychiatrists, and judges [26].

Last but not least important, both victim groups had a comparable percentage of screening tests for STDs performed, suggesting the increased adherence of clinicians to the institutional protocol for clinical assessment in cases of childhood sexual abuse. This trend could be due to either the important legal consequences associated with this kind of offence occurring in children, and to the multidisciplinary team typically involved in this situation.

As expected, a positive screening test for STDs was significantly more present among adult victims than minors. Additionally, more than half of the women with genital lesions had a positive screening test for STDs, suggesting a possible relationship between the two factors. This means that clinicians should systematically suggest that victims presenting with genital injuries have screening tests for STDs and consume the appropriate prophylaxes.

### 4.3. Clinical Implications

The main implication of this retrospective study is that it provides a picture of the differences and clinical findings in cases of SV against women and girls in reference to a single university hospital. Given that there is a limited availability of trained and sensitized personnel in the health workforce globally, a deeper knowledge of the problem may be part of the strategy to appropriately respond to female SV. In particular, training clinicians to evaluate children for suspected abuse should be a priority. According to research, programs that function in isolation, with a low patient volume and inexperienced staff, experience a high degree of error in diagnosis. Trying to reduce errors in diagnosis should be of crucial importance, since they could lead to unnecessary investigations, legal consequences and psychological trauma to the child and family [27]. Specifically, in our setting, there is the need to periodically train the medical and nursing staff to appropriately manage victims of SV. Since the reliable interpretation of the medical findings is one of the most controversial aspects in forensic medical practice, we suggest that a medico-legal specialist is available in the team caring for the sexually abused victim. Moreover, the staff should be informed using the table recently updated by Kellogg and colleagues on the interpretation of medical findings in suspected child sexual abuse [22]. On the other hand, the main clinical implication involving SV in adult women in our case series may be the need to better cooperate with psychiatric specialists in order to decrease the patient’s vulnerability.

Finally, we believe that our findings support academic research on preventive strategies for SV. Collecting research data is essential to guide and structure actions against SV that reflect major social issues, such as fighting inequalities, enhancing sexual health, accessing education and securing familial environments [28].

### 4.4. Strengths and Limitations and Research Implications

Several weaknesses of the study should be acknowledged. First, the number of cases collected, whose results were spread over 12 years, is relatively small. This is due to the low population density present in this geographical area. Second, our review of the case series suggested that gynecologists, who mostly performed the clinical examination, made certain mistakes in terminology and were often inaccurate in their description of genital and extragenital injuries [29]. By contrast, the great amount of data retrieved for each patient in our case series should be recognized as the main strength of this study, which also illustrates how the phenomenon occurred over the last 12 years in our geographical area. Furthermore, the quality of data improved over time after the last update of the institutional protocol transposing the Italian guidelines on the response against female sexual abuse were published in November 2017 and updated in 2020 [10,30]. Of note, understanding the characteristics of the problem and analyzing the healthcare response in different local contexts not only facilitates greater awareness among healthcare professionals, but also serves as a valuable tool in directing skills and resources within the specific territorial scope to ensure the most accurate patient care. Subsequently, to obtain a comprehensive, national, global view of the issue, studies similar to ours could be conducted in various provinces and regions of Italy with the aim of enhancing the identification and management processes of cases of sexual abuse.

Sexual violence is a strongly gendered problem. Therefore, the vast majority of those who are victimized are women, but also trans-identified and non-binary people. Future research should focus on educational measures towards gender-based violence prevention, such as education on the so-called by-stander prosocial behavior [31]. Encouraging bystanders to take positive action not only serves as a preventive measure, but also fosters a culture that prioritizes respect, empathy and equality. The ultimate goal of such an educational strategy should be the eradication of the harmful aspects of masculine culture, thereby cultivating a community where gender-based violence becomes inconceivable.

## 5. Conclusions

In conclusion, this retrospective study exploring SV against women and girls around Udine, in North-East Italy, shows that women under 30 years are the most frequent victims and that, compared with minors, adults more often have bodily injuries and risk factors such as mental health disorders and past experience of SV. By contrast, minors tended to delay seeking medical help, confirming that young people face a number of different barriers such as limited support and feelings of self-blame, shame and guilt, when choosing to disclose.

## Figures and Tables

**Table 1 ijerph-21-00301-t001:** Socio-demographical characteristics and risk factors for SV of all victims, and comparison between minors (<18 years) and adult victims.

	General Population	<18 Years	>18 Years	*p*-Value
Number of cases	155	40/155 (25.8%)	115/155 (74.1%)	
Mean Age (years) ± SD	28.48 (±15.36)	10.8 (±4.94)	34.6 (±12.8)	
Socio-demographical characteristics of SV victims				
Italian	95/155 (61.2%)	27/40 (67.5%)	68/115 (59.1%)	ns
Eastern European	31/155 (20.0%)	4/40 (10.0%)	27/115 (23.4%)	ns
Western European	3/155 (1.9%)	2/40 (5.0%)	1/115 (0.8%)	ns
Asian	3/155 (1.9%)	1/40 (2.5%)	2/115 (1.7%)	ns
Central-South American	9/155 (5.8%)	1/40 (2.5%)	8/115 (6.9%)	ns
African	13/155 (8.3%)	5/40 (12.5%)	8/115 (6.9%)	ns
Missing information	1/155 (0.6%)	0/40 (0%)	1/115 (0.8%)	ns
Single	123/155 (79.3%)	40/40 (100%)	83/115 (72.1%)	<0.0001
Women with a current partner	30/155 (19.3%)	0/40 (0%)	30/115 (26.0%)	<0.0001
Missing information (riguardo single/non single)	2/155 (1.2%)	0/40 (0%)	2/115 (1.7%)	ns
Divorced/separated among single women	20/155 (12.9%)			
Divorced/separated among women with a current partner	4/155 (2.5%)			
Pre-nursery	11/155 (7.0%)	11/40 (27.5%)	-	-
Student	42/155 (27.0%)	26/40 (65.0%)	16/115 (13.9%)	<0.0001
Employed	59/155 (38.0%)	1/40 (2.5%)	58/115 (50.4%)	<0.0001
Unemployed	38/155 (24.5%)	2/40 (5.0%)	36/115 (31.3%)	0.0005
Retired	2/155 (1.2%)	-	2/115 (1.7%)	-
Missing information	3/155 (1.9%)	0/40 (0%)	3/115 (2.6%)	ns
Risk factors for sexual violence				
Previous history of SV	74/155 (47.7%)	12/40 (30.0%)	62/115 (53.9%)	0.010
Previous history of alcohol or drug abuse	13/155 (8.3%)	1/40 (2.5%)	12/115 (10.4%)	ns
Mental health disorder	31/155 (20.0%)	2/40 (5.0%)	29/115 (25.2%)	0.005
History of prostitution	3/155 (1.9%)	0/40 (0%)	3/115 (2.6%)	ns
No risk factors identified	66/155 (42.5%)	26/40 (65.0%)	40/115 (34.7%)	0.001

Legend: SD: standard deviation. SV: sexual violence. ns: not significant. *p*-value < 0.00001: highly significant

**Table 2 ijerph-21-00301-t002:** Characteristics of sexual abuse and the reported gynecological or physical injuries of the minor group versus the adult group.

	<18 Years (*n* 40)	>18 Years (*n* 115)	*p*-Value
Characteristics of sexual violence			
Time interval between SV and medical help request (days)	14.07 (±19.95)	3.7 (DS 10.59)	0.006
Sexual Violence according to WHO definition, committed by one aggressor	40/40 (100%)	105/115 (91.3%)	ns
Group SV	0/40 (0%)	10/115 (8.6%)	ns
Physical violence (added to SV)	0/40 (0%)	20/115 (17.3%)	0.0021
No penetrative SV disclosed (no contact with genitalia has occurred)	14/40 (35.0%)	6/115 (5.2%)	<0.0001
Genital penetration could not be disclosed by the victim	6/40 (15.0%)	32/115 (27.8%)	ns
Victims threatened with a cold weapon	0/40 (0%)	9/115 (7.8%)	ns
Kidnapping	0/40 (0%)	10/115 (8.6%)	ns
Victim showing alcohol/drug abuse before suspected SV	7/40 (17.5%)	31/115 (26.9%)	ns
Toxicological screening tests performed	4/40 (10.0%)	35/115 (30.4%)	0.010
Positive toxicological screening tests	4/4 (100%)	19/35 (54.2%)	ns
Negative toxicological screening tests	0/4 (0%)	16/35 (45.7%)	ns
Positive toxicological screening tests among victims who do not remember SV (victims < 18 years: 3, 2 tests performed; victims > 18 years: 34, 15 tests performed)	2/2 (100%)	10/15 (66.6%)	ns
Suspected use of drug rape	1/40 (2.5%)	4/115 (3.4%)	ns
Known aggressor	29/40 (72.5%)	73/115 (63.4%)	ns
Unknown aggressor	11/40 (27.5%)	42/115 (36.5%)	ns
Gynecological evaluation			
Genital injuries	10/40 (25.0%)	16/115 (13.9%)	ns
Extragenital injuries	5/40 (12.5%)	59/115 (51.3%)	<0.0001
No injuries detected	26/40 (65.0%)	43/115 (37.3%)	0.003
Sexually transmitted screening test performed	35/40 (87.5%)	103/115 (89.5%)	ns
Positive STD screening test	4/35 (11.4%)	59/103 (57.2%)	<0.0001
Negative STD screening test	31/35 (88.5%)	44/103 (42.7%)	<0.0001
Positive HPV infection test	2/4 (50%)	47/59 (79.6%)	ns
Positive HCV infection test	0/4 (0%)	3/59 (5.0%)	ns
Positive HSV1–2 infection test	1/4 (25.0%)	5/59 (8.4%)	ns
Positive Chlamydia infection test	1/4 (25.0%)	6/59 (10.1%)	ns
Positive STD screening tests among victims with genital injuries (tot: 23; victims < 18 yrs: 8 tests; victims > 18 yrs: 15 tests)	0/8 (0%)	8/15 (53.3%)	0.019
Spermatozoa research tests performed	18/40 (45.0%)	83/115 (72.1%)	0.003
Positive spermatozoa research tests	4/18 (22.2%)	24/83 (28.9%)	ns
Negative spermatozoa research tests	14/18 (77.7%)	58/83 (69.8%)	ns
Not evaluable spermatozoa research test	0/18 (0%)	1/83	ns
Positive spermatozoa research tests among victims with genital injuries (tot: 17; < 18 yrs: 5; > 18 yrs: 12 test performed)	0/5 (0%)	3/12 (25.0%)	ns
STD prophylaxis	12/40 (30%)	76/115 (66.0%)	<0.0001
Use of emergency contraception	4/40 (10.0%)	49/115 (42.6%)	<0.0001
Medical therapy refusal	0/40 (0%)	8/115 (6.9%)	ns
Victim follow up			
Psychological–Psychiatric consultation	22/40 (55.0%)	20/115 (17.3%)	<0.0001
Social worker follow up	8/40 (20.0%)	8/115 (6.9%)	0.031
Victim discharge			
Discharge home	18/40 (45.0%)	60/115 (52.1%)	ns
Discharge to emergency room	13/40 (32.5%)	29/115 (25.2%)	ns
Admission to other hospital’s wards (for example psychiatric or pediatric clinic)	5/40 (12.5%)	8/115 (6.9%)	ns
Discharge at anti-violence center	1/40 (2.5%)	4/115 (3.4%)	ns
Victim in custody with police officers	0/40 (0%)	2/115 (1.7%)	ns
Other (for example the victim left the hospital or went back to previous community)	1/40 (2.5%)	5/115 (4.3%)	ns
Missing information	2/40 (5.0%)	7/115 (6.0%)	ns
Mean duration of medical service (hours)	1.772	3.154	<0.0001

Legend: SV: sexual violence. STDs: sexually transmitted diseases. HPV: Human Papillomavirus. HCV: Hepatitis C virus. HSV1-2: Herpes Simplex virus 1-2. ns: not significant

## Data Availability

The data presented in this study are available upon request from the corresponding author.

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
