# Peer review of "Female Sexual Violence: A 12-Year Experience at a Single University Hospital in North-East Italy"

_ijerph, 2024, doi:10.3390/ijerph21030301_

Round 1
Reviewer 1 Report (Previous Reviewer 3)
Comments and Suggestions for Authors.
Author Response
We would like to thank the Reviewer for the thoughtful comments and efforts towards improving our manuscript.
Reviewer 2 Report (Previous Reviewer 4)
Comments and Suggestions for Authors
Dear authors,
The new version of the manuscript includes notable improvements in both structure and content. As far as my comments are concerned, I would like to say that almost all suggestions have been addressed:
· The presentation of the abstract has been changed, data (e.g., percentages and p-values) have been removed and reworded.
· Text has been expanded both in the introduction and in the discussion.
· Two new sections have been added: clinical implications and strengths and limitations.
o One minor issue: in the Clinical implications section, we recommend that the authors include the reference number of Kellogg and colleagues (line 538).
· A major change has been made to the presentation of the tables in the Results section.
· The references section has been expanded and updated.
In short, in our opinion, the "new version" of the manuscript has overcome the weaknesses of the initial version and, consequently, deserves our positive evaluation.
Congratulations!
Author Response
We are glad that this new version of the manuscript satisfies the Reviewer's requirements. We also would like to thank the Reviewer for his/her suggestions and recommendations. We recognize that the manuscript sounds much better now.
Regarding the minor concern raised, specifically the suggestion to incorporate the reference number for Kellogg and colleagues in the "clinical implication section," we wish to affirm to the Reviewer that we have duly addressed this recommendation (see line 355 in the new version of the manuscript)
Reviewer 3 Report (Previous Reviewer 5)
Comments and Suggestions for Authors
The manuscript attempts to address the management of female SV victims in the emergency room. This goal has been achieved by the authors. Congratulations.
We kindly remind the authors that all scientific work has an educational and pedagogical part that may raise limitations and possible future actions. This is where we kindly ask the authors to propose the prosocial action of bystanders as a preventive and pedagogical measure. This offers a promising alternative to mitigate the incidence of gender-based violence. We therefore recommend the authors to further explore the importance of educational interventions to increase prosociality against gender-based violence among bystanders. Remember that in the issue of gender-based violence, any preventive measure, especially in the educational and preventive field, is very important and necessary. We suggest the following document: DOI: 10.3390/socsci12070406
Author Response
We would like to thank the Reviewer for the kind suggestion. We further improved the section "Research implication" adding the following paragraph (lines 376-381 in the latest version of the manuscript):
"Sexual violence is a strongly gendered problem. Therefore, the vast majority of those who are victimized are women, but also trans-identified and non-binary people. Future research should focus on educational measures towards gender-based violence prevention, such as the so-called by-stander prosocial behavior [31]. Encouraging bystanders to take positive action not only serves as a preventive measure but also fosters a culture that prioritizes respect, empathy and equality. The ultimate goal of such educational strategy should be the eradication of harmful aspects of masculine culture, thereby cultivating a community where gender-based violence becomes inconceivable."
We also added the reference suggested by the Reviewer. We hope to have satisfied the Reviewer's requirements.
This manuscript is a resubmission of an earlier submission. The following is a list of the peer review reports and author responses from that submission.
Round 1
Reviewer 1 Report
Comments and Suggestions for Authors
Dear authors, thank you for the opportunity to evaluate the article that presents such relevant research.
Although the research is extremely important and provides relevant data to better serve victims of violence in the health system, there are points that can be improved. This will make the text more reader-friendly and eliminate doubtful inductions regarding the understanding of the phenomenon studied.
1. It is suggested to provide another format for written presentation and discussion of the content of the tables. As it is, reading becomes difficult and very dense.
2. It is extremely important to add to the references studies on the underreporting of violence suffered by minors under 18 and the factors that make such reporting difficult.
3. It will also be important to discuss these factors in the discussion and highlight them in the conclusions to eliminate misunderstandings.
4. It will be important to expand the conclusions.
Reviewer 2 Report
Comments and Suggestions for Authors
1) It seems the authors have used a mixed methods approach to the research based on the findings presented. However, in the "Materials and Methods" section the authors only state statistical (Quantitative) methods used. Could the authors please clarify their methodological stance? Thank you.
2) Clinical implications section needs to be vastly improved based on the high amount and quality of statistical data presented in the paper. Suggestions include:
-How may this localized study be applied to a national context?
-How may a trained and sensitized workforce be developed in the geographical context in which the study was conducted?
-Based on the findings, what would more appropriate responses to violence against women be for trained clinicians?
-Please elaborate further on the clinical implications of the differences in sexual violence between adults and minors. At present this is not fully expounded. By expounding further the authors would be able to make full use of the data and place their paper in a more solid academic foundation.
Comments on the Quality of English LanguageMinor corrections needed.
Reviewer 3 Report
Comments and Suggestions for Authors
Dear authors,
I thank you for your work which seems really interesting and offers food for thought both for the field of scientific research and for social policies in favor of victims of sexual violence, and the prevention of such criminal actions.
I think the work is really interesting and conducted with methodological correctness. The paper is packed with information and the authors offer a discussion of the interesting findings. The sample is small, but adequate for the analyzes and objectives of the research.
I just have a few pointers:
- extend the arguments regarding the prevalence of forms of sexual abuse against girls and women, with particular reference to the Italian context and, where possible, to the regional context.
- where the 6 cases of childhood sexual abuse are described, I think that a mention of the issue of the credibility of minors and their impact on child protection could find the right space (see Longobardi et al, 2023).
Suggested References (not mandatary)
Longobardi, C., Malacrea, M., Giulini, P., Settanni, M., & Fabris, M. A. (2022). How plausible are the accounts of child victims of sexual abuse? A study of bizarre and unusual scripts reported by children. Journal of child sexual abuse, 31(2), 216-235.
Reviewer 4 Report
Comments and Suggestions for Authors
The authors' interest in illustrating the magnitude of the female sexual violence is of great interest to me. Furthermore, the analysis of the risk factors, as well as making comparisons according to age (115 adult victims and 40 minors) is, in my opinion, a step forward in the development of effective prevention and intervention programs for this problem. I would therefore like to express my appreciation for their work.
In an effort to contribute, as a reviewer, to pointing out some aspects that will probably contribute to improving the quality of the work, I would like to make some recommendations:
Abstract
· I do not consider it necessary to include the words "Background", "Result", "Methods", "Conclusions" as they do not help the reading.
· So much data (percentages and p-values) do not help the potential reader either. Here it would be enough to point out the trends resulting from the comparisons (the concrete data are already in the results section).
· I do miss the objective(s) of the study.
Introduction
· I think this section should be worked on a little more. A better and more detailed presentation of the status quo of the field is needed to support and substantiate the objective of the study. These are some aspects I advise to include from a solid review of the previous literature:
o Some data on the nature and extent of violence against women in different geographical settings.
o Some consideration, from the literature, of the different types of violence against women and the difficulties inherent in coming up with a definition of violence against women.
o Indicate, on the basis of previous literature, what are the potential risk factors for violence and the multiple negative consequences.
o In short, the aim is to provide the context to support the "need for this study. Although the authors point out that in Italy there has been an increase in the rate of gender-based violence, there is no answer to the question: what does this study advance with respect to previous literature?
o Although there is a general objective, it would be necessary to refer to their interest in comparing the characteristics of sexual violence against women in the two age groups established.
Material and methods
· In my opinion, this section would be clearer if the authors structured it in different sub-sections: Procedure, participants, study variables and statistical analysis.
· It would be advisable to refer to the socio-demographic characteristics of the total sample (which appear in the first part of Table 1) in the subsection on participants. The analysis of the differences in these variables between the two age groups could be included in the results section.
· In describing the variables under analysis in this study, could the authors categorise them in some way? (in the presentation of the tables, the authors group them under certain headings)
Results
· I consider it unnecessary to include the term "Tables" in section 3.1. Results and Tables.
· In order to facilitate the presentation of the results, I suggest to the authors a better layout of the tables, as well as some changes in the presentation of the content of the tables.
o Table 1 would include the data corresponding to the socio-demographic characteristics of the total sample and of the two groups established according to age; the table would also provide the p-values for comparison between groups.
o Table 2 would present the results of the Risk factors for sexual violence, also for the total sample and the two groups, as well as the p-values of the between-group comparison.
o Table 3 would present the characteristics of sexual abuse and the reported gynecological or physical injuries for the total sample and the two groups, as well as the p-values for the between-group comparison.
· Table 4 has the same content as Table 3. The authors should correct the error and present Table 4 with data describing "the characteristics of sexual abuse and the reported gynecological or physical injuries of the minors group versus the adults group".
· I advise the authors to revise the labels of the various tables and rewrite them to indicate what they include. For example, instead of "Table 1 describes the socio-demographical characteristics and risk factors for SV...." it should be Table 1. Socio-demographical characteristics and risk factors for SV....
Discussion
· In general, the reading of this section is not particularly clear. Some suggestions for authors:
o Remove the two sub-sections of Principal findings and Results in the Context of What is Known.
o Discuss in more depth, taking into account the review by Lemaigre, Taylor & Gitto (2017), of the finding that "victims younger than 18 years tended to delay medical help seeking". Can the authors advance some of the factors that prevent or facilitate seeking medical help in this type of situation? We consider these to be important questions for preventive and interventional measures for children and young people.
o Include clinical implications as well as strengths and limitations in sections other than the discussion.
o Elaborate some content regarding the implications of this study for future research.
o To expand on the practical implications derived from this study, advancing, along the lines of other studies, some specific guidelines for action in the prevention and/or intervention of this problem in the socio-cultural context analyzed…
References
· Although the authors include articles published in prestigious journals, we suggest a more extensive review of the extensive literature on this topic (with a special emphasis on recent work).
Reviewer 5 Report
Comments and Suggestions for Authors
The manuscript deals with Female sexual violence: 12 years of experience in a university hospital in North-Eastern Italy.
The study found the following results: Compared to minors, adults presented risk factors such as mental health illness and previous experience of sexual violence and reported bodily and extragenital injuries more frequently. In addition, positive screening for sexually transmitted diseases and their association with genital injuries was found to be significantly more present among adult victims than among minors. In contrast, victims under 18 years of age tended to delay seeking medical help and more often did not report genital penetration.
This study sheds light on how a deeper understanding of the magnitude of the problem in a specific geographic area can be part of the strategy to respond appropriately to female sexual violence.
The introduction and discussion need to be strengthened, so I indicate below some elements that the authors need to improve.
The authors should provide statistical data from Europe and globally on the number of women requiring medical help due to violence in the period 2014-2019, before the pandemic outbreak. This can be used to compare whether the increase in Italy and in Europe or the world are similar or different.
Given that gender-based violence represents a problem of public interest with high prevalence, this has intensified the preventive strategy for potential victims. However, prosocial bystander action provides a promising alternative to mitigate its incidence.
We recommend the authors to further explore the importance of educational interventions to increase prosociality against gender-based violence in bystanders.
The method is well described.
In the discussion, the suggestions I have included to strengthen the introduction should also be taken into account.
Proposals for the future should be pointed out.
The conclusions are convincing with the evidence and arguments presented.
The citations and references are appropriate and up to date, but the authors should review the journal's rules, so that they are as requested by the International Journal of Environmental Research and Public Health.
Tables are very clear, but should be checked to ensure that they are in the format requested by the journal in which they are to be published.
The data are reproducible.
The authors contribute knowledge to science with this manuscript, being of relevance and interest.